# Liver Fat Storage Is a Better Predictor of Coronary Artery Disease than Visceral Fat

**DOI:** 10.3390/metabo13080896

**Published:** 2023-07-28

**Authors:** Maamoun Basheer, Elias Saad, Helena Jeries, Nimer Assy

**Affiliations:** 1Internal Medicine Department, Galilee Medical Center, Nahariya 221001, Israel; maamon.basheer@mail.huji.ac.il (M.B.); eliass@gmc.gov.il (E.S.); mymonbasher@gmail.com (H.J.); 2Azrieli Faculty of Medicine, Bar-Ilan University, Safad 1311502, Israel

**Keywords:** fatty liver, visceral fat, atherosclerosis, coronary plaques, cardiac CT, biomarkers, carotid intima–media thickening

## Abstract

Fatty liver is one aspect of metabolic syndrome. The roles and contributions of fatty liver and visceral fat storage to coronary artery disease (CAD) are not clear. This study measured associations among visceral fat storage, fatty liver, insulin resistance, atherosclerosis, and CAD. Patients were divided into three groups: excess visceral fat (visceral fat area >330 ± 99 cm^2^), non-alcoholic fatty liver disease (NAFLD), and a control group. The definition of fatty liver is liver minus spleen density greater than or equal to −10. We defined early atherosclerosis as intima–media thickness of the common carotid artery >7 mm in men and >0.65 mm in women, measured with Doppler ultrasound. Visceral fat area was defined using CT (>330 ± 99 cm^2^). Insulin-resistance biomarkers (HOMA), CRP, and oxidant–antioxidant status (MDA-Paraoxonase) were also measured. Patients with high liver or visceral fat showed higher coronary plaque prevalence (50% (*p* < 0.001), 38% (*p* < 0.01), respectively vs. 25% in the control group), higher prevalence of coronary stenosis (30% (*p* < 0.001), 22% (*p* < 0.01) vs. 11% in the control group), higher intimal thickening (0.98 ± 0.3 (*p*< 0.01), 0.86 ± 0.1 (*p* < 0.01) vs. 0.83 ± 0.1 in the control group), higher HOMA (4.0 ± 3.0 (*p* < 0.005), 3.0 ± 1.0 (*p* < 0.001) vs. 1.5 ± 1.2 in the control group), and higher triglyceride levels (196.8 ± 103 (*p* < 0.005), 182.6 ± 90.87 (*p* < 0.005) vs. 145 ± 60 in the control group). Multiple logistic regression analysis showed that fatty liver predicted CAD (OR 2.7, 95% CI 2.3–4.9, *p* < 0.001) independently of visceral fat storage (OR 2.01, 95% CI 1.2–2.8, *p* < 0.001). Liver fat storage is a strong independent risk factor for CAD and carotid atherosclerosis and contributes more than visceral fat storage.

## 1. Introduction

Obesity is a worldwide epidemic [1], and body mass index (BMI) or waist circumference is mostly used to quantify overall adiposity. However, fat deposits may be more important than overall adiposity [2,3]. Adipose tissue can be morphologically classified into white, brown, and beige subsets [1,2,3,4,5,6]. White adipose tissue is broadly classified by location, largely defined as subcutaneous (located under the skin) and visceral/omental (located intra-abdominally, adjacent to internal organs). Adipose tissue comprises many different cell types, which coordinately secrete numerous cytokines, chemokines, and hormones [1,2,3,4,5,6]. Ethnicity also affects fat distribution. Asians and Blacks appear to have more central fat, upper body fat, and subcutaneous and visceral abdominal adipose tissue than Caucasians. However, the ethnic difference in fat distribution is less well understood [4,5].

Abdominal visceral adipose tissue (VAT), which includes the largest visceral fat depots in the body, significantly correlates with cardiovascular disease and represents a high risk of metabolic syndrome [4,5,6]. VAT releases adipokines and free fatty acids (FFAs) directly into the portal vein, through which it is delivered to the liver [6,7,8]. This has caused researchers to theorize VAT′s systemic effect on atherosclerosis. Non-alcoholic fatty liver disease (NAFLD), also known as metabolic dysfunction-associated fatty liver disease, is when there is excessive liver fatty build-up without other causes, such as alcohol. Chronic metabolic diseases such as obesity and type 2 diabetes are strong NAFLD risk factors [8]. Type 2 diabetes mellitus and insulin resistance facilitate lipolysis in adipose tissue. This leads to the release of free fatty acids from their storage in the liver. This leads to the development of hepatic steatosis [9].

NAFLD diagnosis is based on the presence of hepatic steatosis and one of the following three criteria: overweight/obesity, type 2 diabetes mellitus diagnosis, or having evidence of metabolic dysregulation [10]. Liver biopsy is considered the gold standard for NAFLD diagnosis and staging. However, the majority of patients can be effectively non-invasively diagnosed with routinely available tests. Imaging techniques for diagnosing and assessing liver steatosis are also being developed to overcome the risks of invasive procedures. Several methods have been shown to warrant future evaluation, such as ultrasonography, elastography, magnetic resonance imaging, magnetic resonance spectroscopy, computed tomography, and chemical shift imaging [11].

People with NAFLD mostly die from cardiovascular diseases [12,13,14]. Although cardiovascular disease is the leading cause of mortality in patients with NAFLD, the association of coronary heart disease and NAFLD is affected by various diseases, like diabetes, and having a high BMI [12,13,14]. Some studies emphasize the correlation among fatty liver disease, visceral fat storage, and metabolic syndrome, a condition associated with a high risk of coronary artery disease (CAD) [15,16,17,18]. Therefore, every patient suffering from NAFLD should be assessed for cardiovascular disease. However, there is a lack of consensus on patient selection for screening. Additionally, the preferred diagnostic method remains unclear [15].

The individual contribution of visceral and liver fat storage to the risk of CAD also remains unclear. Therefore, the study’s aim is to assess the association among liver and visceral fat storage, insulin resistance, carotid atherosclerosis, and coronary artery diseases.

## 2. Materials and Methods

### 2.1. Study Population 

This was a prospective study that included 190 obese patients. The patients were referred to the hospital for cardiac CT as part of routine clinical care. Electronic medical records (EMRs) were collected.

### 2.2. Eligibility Criteria

#### 2.2.1. Inclusion Criteria

Obese males or females between 30 and 75 years old, with low to intermediate risk of CAD and without other liver or biliary disorders.

#### 2.2.2. Exclusion Criteria

BMI > 35.Acute illness.Previous ischemic heart disease or cerebrovascular disease.Patients suffered from typical chest pain.Previous CAD, conventional coronary angiography, coronary bypass grafting, renal failure, percutaneous intervention, or cancer.Hepatic steatosis agents like steroids, estrogens, methotrexate, and procor.Cardiac CT exclusions: presence of multiple ectopic beats, heart rate of more than 75/min despite therapy, atrial fibrillation, severe lung disease, and a history of allergic reaction to iodine-containing contrast agents.

### 2.3. Study Design 

A total of 190 obese patients of different ethnicities (Jewish and Arab) were referred to the hospital for a cardiac CT. In total, 90 patients were excluded due to high risk of CAD (52 with percutaneous coronary intervention and 38 with coronary artery bypass graft). The remaining 100 patients were at low risk of CAD and were selected for study recruitment (Figure 1). 

Patients with high visceral fat or diffuse liver fat were divided into two groups: Group 1 included 70 patients with large visceral fat area (330 ± 99 cm^2^), and Group 2 included 30 patients with diffuse fatty liver (liver–spleen density ≥ −10 HU). The third group, containing 30 individuals, matched according to age, gender, and BMI, served as a control group.

All subjects underwent complete family history analysis, BMI measurements, and a physical examination. They were also evaluated for insulin-resistance markers (fasting glucose, HOMA). Insulin resistance was estimated using the homeostasis model assessment (HOMA-IR) derived from the following equation: 

Insulin resistance (IR) = ((fasting plasma glucose level mg% × 0.055) × (fasting plasma insulin level mU/L)/22.5) (as previously described [14]). 

Obesity was defined if the patient′s calculated BMI was more than 28 kg/m^2^ but less than 35 kg/m^2^, and diabetes was determined as HA1C > 6.5% or fasting plasma glucose levels >126 mg/dL. Metabolic syndrome was defined as previously described [15]. Triglyceride and cholesterol levels were measured. Other parameters were also measured. Inflammation markers (CRP and fibrinogen), markers of oxidant–antioxidant status (MDA, paraoxonase, and alpha-tocopherol), and paraoxonase activity were measured according to previously described methods [19,20]. The metabolite α-tocopherol was estimated spectrophotometrically [19,20]. Lipid peroxidation (MDA concentration) was estimated spectrophotometrically with the use of a thiobarbituric acid assay [21,22]. 

### 2.4. Hepatic Steatosis

Hepatic steatosis diagnosis was defined as attenuation of at least –10 HU (calculated as liver attenuation minus spleen attenuation) detected using CT [23]. The measurement of liver and spleen attenuation was performed in a blinded way. Hepatic attenuation was measured by randomly selecting three circular regions of interest (ROI) on three transverse sections at different hepatic levels. The ROI values were averaged as the mean hepatic attenuation. To provide an internal control, the mean splenic attenuation was calculated by also randomly averaging three ROI values of splenic attenuation on three transverse sections at different splenic levels (Figure 2). 

NAFLD diagnosis was made relying on the radiologic findings of hepatic steatosis along with < 20 g/day alcohol use, negative hepatitis B or C virus serology, negative autoantibodies for autoimmune hepatitis, and no known history of any other known liver disease. 

### 2.5. Assessment of Carotid Atherosclerosis

All patients underwent carotid ultrasounds for intima–media thickness (IMT) evaluation and plaques. A high-resolution ultrasound device (Sonos 5500; Agilent Technologies, Palo Alto, CA, USA) and a 10 MHz linear array transducer were used for carotid scans. Longitudinal views of the following were obtained: the left and right common carotids, carotid bifurcations, and internal and external carotid arteries. All ultrasound studies (carotid and brachial flow-mediated dilation) were analyzed off-line using the Prosound system by specially trained technicians who were blinded to other study variables. IMT was measured over 10 mm in the far wall of the common carotid within 2 cm proximal to the bulb (Figure 3). 

The thickest IMT region without focal lesions was measured. Both right and left carotids’ IMT values were averaged. The intima–media area was defined as previously described [24]: 

Intima–media area in mm^2^ = (IMT mm) × (length over which IMT was measured in mm).

Plaques were quantified for all the carotid segments (common, internal, and external carotid arteries). Normal ranges were <7 mm for males and <0.65 for females. 

### 2.6. Assessment of Visceral Fat 

The content of abdominal adipose tissue was measured with CT scans at the L4–L5 vertebra level. Subjects were in the supine position with both arms stretched above their heads. A single 6 mm slice was taken during respiration after a normal expiration. The total abdominal adipose tissue (TAT) area was measured and computed using an attenuation range of −190 to −30 HU. The quantification of visceral abdominal fat (VAT) was performed by delineating the abdominal cavity at the abdominal wall’s internal aspect and the vertebral body’s posterior aspect. Superficial adipose fat (SAT) was calculated as VAT area minus TAT area (Figure 4). VAT and SAT volumes were obtained by multiplying the area for each fat component by the slice thickness [25].

### 2.7. Assessment of Coronary Atherosclerosis 

Identifying coronary atherosclerosis was performed using a cardiac CT protocol. A bolus of 70–90 mL iomeprol (400 mg/mL; Iomeron 400; Bracco, Milan, Italy) contrast medium was injected intravenously (4 mL/s) followed by a 50 cc bolus of saline, placed in the antecubital vein with an 18-gauge catheter. The scan delay was determined using an automatic bolus test in which the ROI was located on the ascending aorta. Patients were instructed to maintain an inspiratory breath hold while acquiring both CT data and electrocardiogram (ECG) traces. A 64-row scanner with slice thickness of 0.625 mm was employed. Beta blockers (Propranolol, 20 mg tab) were given orally if there was >70 beats per minute (BPM) during resting heartrate. If the heartrate was >80 BPM, the patient was excluded from the study. Raw image datasets from all acquisitions were analyzed by two independent specialists (a cardiologist (A.M.) and a radiologist (L.A.), both with over 20 years of experience) who were unaware of the hepatic and lipid profiles of the patients. The two specialists visually graded the degree of stenosis and plaque composition. The stenosis was considered significant if over 50% occlusion of the arterial lumen was present [25,26]. Coronary artery Brilliance Philips Medical Systems, USA disease was defined as a stenosis of over 50% in at least one major coronary artery (Figure 5). 

### 2.8. Ethics

This study was approved by Ziv Medical Center′s ethics committee. The EMR database was analyzed in agreement with the ICH guidelines for good clinical practice. Informed consent was obtained from all the patients after they were selected for the study.

### 2.9. Statistics

The variables were displayed as means ± standard deviations (SDs), and the categorical variables were reported as percentages. Chi-square tests or Fisher’s exact tests for categorical variables, and Student’s *t*-tests for continuous variables with normal distributions or Mann-Whitney tests for non-normally distributed variables were used. Correlations between liver fat density, and visceral and subcutaneous fat areas were performed with Spearman′s test. Multiple logistic regression analysis after adjusting for all confounding factors was performed to identify independent predictors of patients having a CAD diagnosis (stenosis of >50% in at least one major coronary artery according to cardiac CT). Candidate variables selected for logistic regression modeling were liver–spleen density > −10, excessive visceral fat (>330 cm^2^), subcutaneous fat > 230 cm^2^, CRP > 2 mg/dL, HOMA > 2.5, and the presence of metabolic syndrome. Covariate variables were age, gender, LDL-cholesterol, and biomarkers. For the final model, Wald statistics and odds ratios (OR) were reported for variables, and the overall model assessed the c-statistics for predicting the diagnosis of CAD. Intra- and inter-observer agreement among readers of cardiac CT and for evaluating liver–spleen density and Kappa statistics were calculated. Statistical analysis was performed using Winstat Statistic for Windows (version 3.0; Kalmia Co. Inc. Southard Dr, Beltsville, MD, USA). 

## 3. Results

### 3.1. Clinical Characteristics of the Study Participants

Of the 190 obese patients who were selected, 90 patients were excluded due to not being matched in quality control distribution according to excessive visceral fat storage or diffuse fatty liver. The remaining 100 patients who could be separately matched according to visceral or liver fat were divided into two groups. Group 1 had large visceral fat area (330–463 cm^2^, liver–spleen density 1.0 ± 7.5 HU), and Group 2 had diffuse fatty liver (liver–spleen density >−20 ± 4.5 HU, visceral fat area 197–330 cm^2^). Thirty healthy obese individuals without high visceral fat (74–230 cm^2^), without diffuse fatty liver (density +7.5 ± 6.9 HU), and matched according to age, gender, and BMI served as controls. 

The clinical and biochemical characteristics of all study participants are summarized in Table 1. The findings of the fatty-liver and the large-visceral-fat-area groups show significant differences in hypertension disease (50% vs. 20%, *p* < 0.05), metabolic syndrome (57% vs. 44%, *p* < 0.05) diabetes (52% vs. 27%, *p* < 0.05), alanine aminotransferases (39 ± 21 vs. 31 ± 17, *p* < 0.05), and triglyceride levels (197 ± 103 vs. 182 ± 90, *p* < 0.05), respectively (Table 1). There were no significant differences between the two groups in age, gender, alcohol intake, BMI, and LDL or HDL serum levels (Table 1). 

### 3.2. Differences in Coronary Plaques, Coronary Stenosis, Carotid Atherosclerosis, Biomarkers of Insulin Resistance, Inflammation, and Oxidative Stress among the Three Groups 

The differences in coronary plaques, coronary stenosis, carotid atherosclerosis, biomarkers of insulin resistance, inflammation, and oxidative stress among the three groups were studied (Table 2). Patients with NAFLD and patients with large visceral fat areas showed higher prevalence of coronary plaques and carotid atherosclerosis than healthy controls. Patients with diffuse fatty liver (Group 2) had significantly more coronary plaques (50% vs. 38%, *p* < 0.001), higher prevalence of coronary stenosis (30% vs. 22%, *p* < 0.001), and greater intima–media thickening (0.98 ± 0.3 vs. 0.86 ± 0.1, *p* < 0.01) than the control group. Higher insulin resistance (HOMA 4.5 ± 3 vs. 3.0 ± 1, *p* < 0.01) and lower MDA serum levels were seen in patients with excessive visceral fat storage (Group 1) compared with the control group (Table 2). There were no significant differences among the groups in terms of MDMA, CRP, and serum paraoxonase levels (Table 2).

### 3.3. Multivariate Logistic Regression Analysis for Predicting Coronary Artery Disease 

We constructed a multivariate model for predicting coronary artery disease. Diffuse fatty liver, excessive visceral fat, subcutaneous fat, CRP, and metabolic syndrome correlated with CAD. Subcutaneous fat storage (>230 cm^2^), CRP (>2 mg/dL), and HOMA (>2.5) had no significant effect on CAD (Table 3).

Diffuse fatty liver was the strongest predictor of CAD (OR 2.75, CI 2.3–4.9, *p* < 0.001) independently of excessive visceral fat (OR 2, CI 1.2–3.8, *p* < 0.001) and of metabolic syndrome (OR 1.52 CI 0.8–3.5, *p* < 0.01) (Table 3). 

## 4. Discussion

NAFLD syndrome is hepatic expression of metabolic syndrome [5,6,7,8,9,10]. It is manifested by insulin resistance and endothelial dysfunction. These manifestations are correlated with the presence diabetes mellitus and obesity [6,7,8]. NAFLD is also associated with hypertension and dyslipidemia [5,6,7,8,9,10]. There are correlations among fatty liver disease, visceral fat storage, and metabolic syndrome, conditions associated with high-risk CAD. We studied the association among liver and visceral fat storage, carotid atherosclerosis, and coronary artery disease [5,6,7,8,9,10]. 

This study shows that patients with either NAFLD or large visceral fat areas had a higher prevalence of coronary soft plaques and coronary stenosis, higher IMT and HOMA, and higher triglyceride levels (Table 2). The main finding of this research is that fatty liver is associated with CAD independently of visceral fat area in obese patients at low to intermediate risk of CAD (Table 2 and Table 3).

Our results are comparable with findings from other studies [27,28,29,30,31,32,33]. Hoenig et al. showed that fatty liver is associated with metabolic syndrome, independently of visceral fat area. However, in Hoenig’s study, cardiac CT was not performed, and intima–media thickening for evaluating atherosclerosis was not measured [27]. Fabrini et al. also showed that intrahepatic triglycerides, not visceral fat, are linked with metabolic complications of obesity [27]. Additionally, the removal of omental fat does not improve cardiovascular risk factors and insulin sensitivity in obese adults [31]. These results emphasize the dangers of fat storage in the liver as compared with visceral fat storage, towards progression to atherosclerosis and coronary artery disease.

Lee et al. [33] showed that visceral obesity is not an independent variable as a metabolic syndrome risk factor. However, fatty liver was significant in this model. They suggested that fatty liver may be a greater risk factor than visceral fat [33]. The metabolic syndrome still the direct factor in development fatty liver and coronary artery diseases [33,34,35,36].

Evidence from recent human and animal studies clearly shows that NAFLD pathogenesis is regulated by both genetic and environmental factors. DNA methylation processes are involved in the prevalence and progression of NAFLD. Many different human studies have demonstrated that altered DNA methylation at global and particular CpG sites is inversely correlated with pertinent gene expression in steatohepatitis patients [37,38,39,40]. However, the direct mechanism by which fatty liver storage contributes to coronary artery disease remains unclear. Possible explanations are dyslipidemia, hypertension, and diabetes, or chronic inflammation-mediated oxidative stress, insulin resistance, subclinical inflammation via increased secretion of IL-6, C-reactive protein and plasminogen activator inhibitor-1, endothelial dysfunction, cytokine upregulation, atherogenic dyslipidemia, postprandial lipemia, pro-coagulation, and hypofibrinolysis [33,34,35,36,37,38,39]. These phenomena mediate oxidative stress and endothelial dysfunction, finally promoting CAD [33,34,35,36,37]. Furthermore, cytokine production is significantly increased when progressing from simple fatty liver to steatohepatitis [37,38,39]. 

FFAs cause lipotoxicity, impair endothelium-dependent vasodilatation, increase oxidative stress, and have cardiotoxic effects [38,39]. In our study, patients with diffuse fatty liver disease had metabolic syndrome, greater insulin resistance, higher triglyceride levels, and greater endothelial dysfunction than those with excessive visceral fat storage (Table 2). Subcutaneous fat storage did not predict CAD, and biomarkers of inflammation and oxidative stress were similar between the two non-control groups. 

Visceral fat includes omental and mesenteric fat. Omental fat has greater potency in metabolic dysfunction [40,41,42,43,44], high lipolysis activity, and high resistance to insulin and has greater gene expression of most proinflammatory adipokines than mesenteric and subcutaneous fat [40,41,42,43,44]. The metabolic function of the liver is controlled by insulin and other metabolic hormones [41,42,43,44]. Fatty acids also affect liver metabolic function [40,41,42]. These fatty acids include long-chain fatty acids, which are incorporated into triacylglycerol, phospholipids, and cholesterol esters in hepatocytes. These complex lipids are stored as lipid droplets or secreted as very-low-density lipoprotein (VLDL) particles into the circulation. In fasting states, lipolysis performed in adipose tissue results in the release of nonesterified fatty acids [41]. 

High concentrations of FFAs inhibit insulin′s ability to stimulate muscle glucose uptake and suppress hepatic glucose production [44,45,46,47,48,49]. This contributes to hepatic insulin resistance [45]. However, only about 20% of total fatty acids delivered to the liver originate from the lipolysis of visceral fat in obese subjects [45,46,47,48]. Therefore, visceral fat is not a major FFA source in either portal or systemic circulations and is thus unlikely to cause insulin resistance via only an FFA-mediated mechanism. This is in accordance with the results of our study (Table 2 and Table 3), in that patients with fatty livers had greater insulin resistance and higher triglyceride levels than patients with excessive visceral fat [49,50,51].

Traditional tools for predicting coronary artery disease, like the coronary calcium score, are extensively used to evaluate and predict future cardiovascular events [49,50,51]. The Framingham risk score, which provides risk stratification for cardiovascular event occurrence within a 10-year time period, is also used [50,51]. Non-traditional tools like fatty liver intensity, as described above, could also help predict and detect subclinical coronary artery disease. The combined use of traditional and non-traditional tools could help evaluate and may guide us in better cardiovascular profiling. 

Our results show that there is a direct correlation between the accumulation of liver fat and atherosclerosis development. Therefore, efforts to decrease liver fat might directly decrease coronary artery diseases. Lipid-profile optimization and improvement in insulin resistance are desirable therapeutic targets in patients with this disease and constitute a substantial research area [52].

One limitation of the study is that the diagnosis of fatty liver was performed with non-enhanced CT rather than liver biopsy. However, a cutoff of liver–spleen density ≥ −10 HU identifies patients with diffuse fatty liver with 87% sensitivity and 90% specificity [23]. Another limitation is that fibro-scans and Fib-4 were not used in this research study, although other accurate tests were performed. The small number of the cohort in this single-center study warrants the further assessment of this issue in multi-center settings and with a wider cohort range. 

## 5. Conclusions

We found that the risk of coronary artery disease was higher in patients with liver fat storage compared with visceral fat storage. This should help optimize cardiovascular risk stratifications. Physicians should not only aggressively treat NAFLD patients for their liver disease but also aggressively treat the underlying CAD risk factors. This is because many patients with NAFLD are likely to have a major CAD event and possibly die prior to developing advanced liver disease. 

## Figures and Tables

**Figure 1 metabolites-13-00896-f001:**
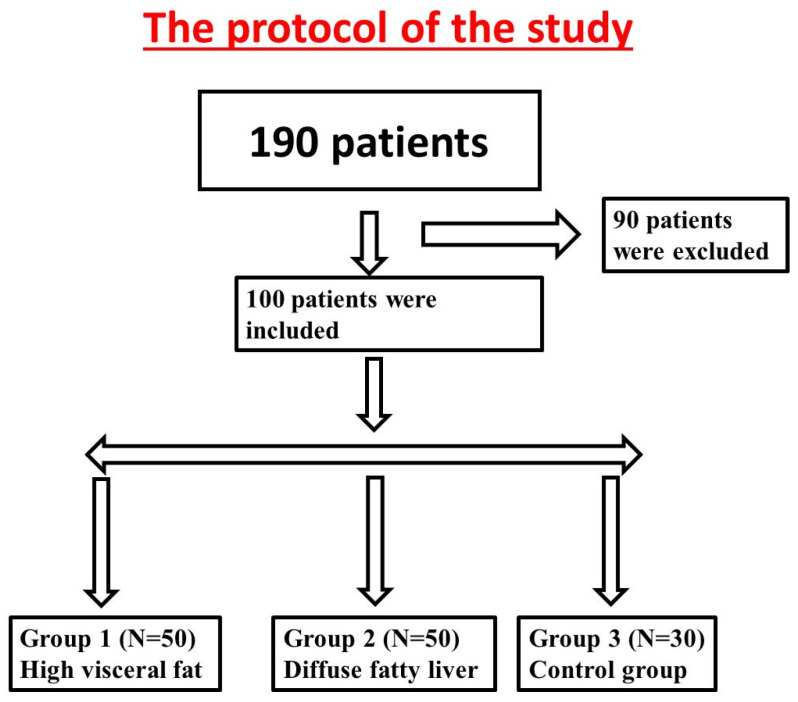
The study design.

**Figure 2 metabolites-13-00896-f002:**
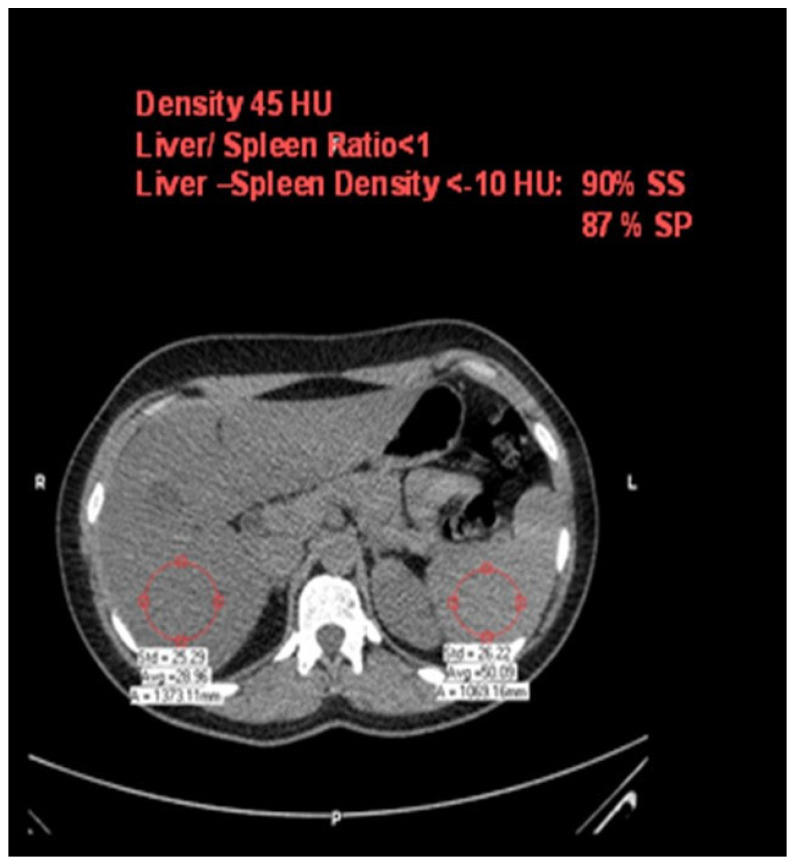
Computed-tomography section showing liver and spleen attenuation.

**Figure 3 metabolites-13-00896-f003:**
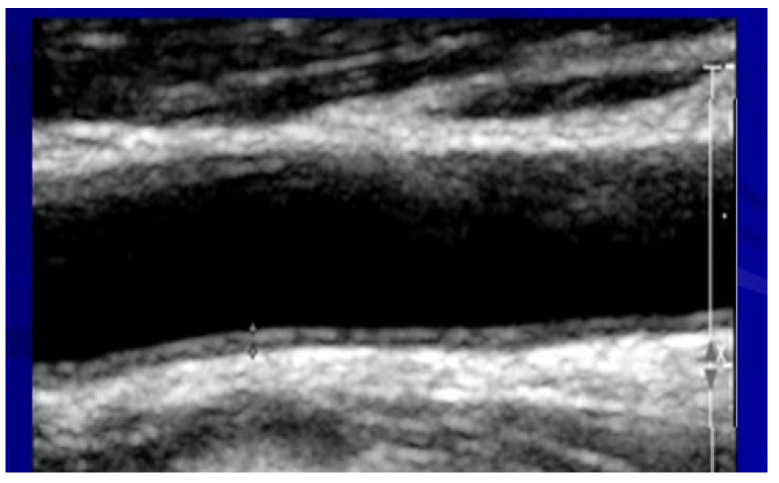
An ultrasound image of the carotid artery for evaluating intima–media thickness.

**Figure 4 metabolites-13-00896-f004:**
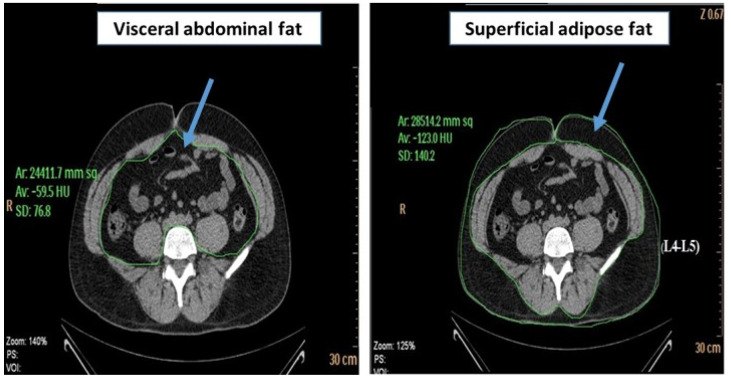
Computed-tomography section showing visceral abdominal fat and superficial adipose fat.

**Figure 5 metabolites-13-00896-f005:**
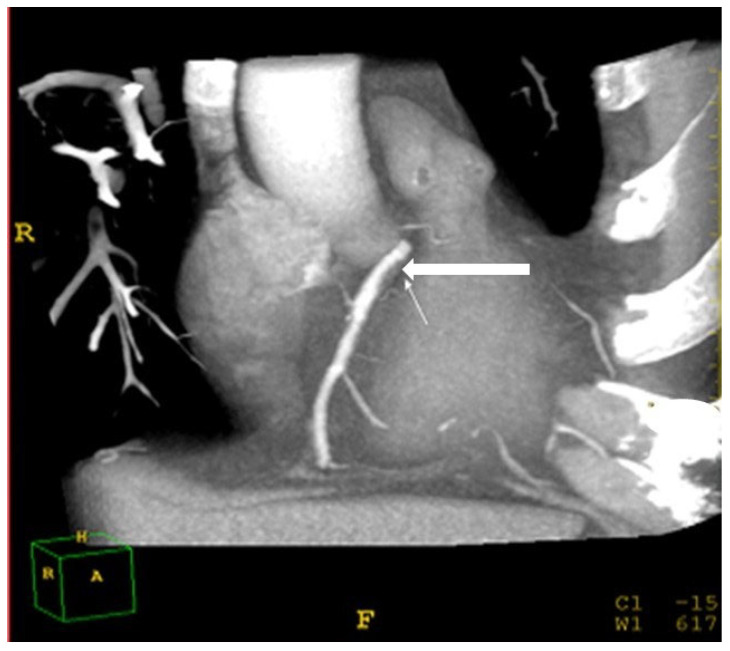
Cardiac computed-tomography section showing a major coronary artery **(white arrow)**.

**Table 1 metabolites-13-00896-t001:** Clinical and biochemical characteristics of all study groups.

	Excessive Visceral Fat Storage N = 30	Fatty Liver Storage N = 70	Controls N = 30	*p*-Value FL vs. Controls
Age (yrs.)	51 ± 8.7	50 ± 9.4	50 ± 10	0.3
Male (% of total)	68	74	75	0.1
Hypertension (% of total)	20	50	24	0.02
Diabetes mellitus (% of total)	27	52	13	0.003
Alcohol intake (g/day)	5 ± 10	7.0 ± 11	5 ± 4	0.3
BMI *	32 ± 5	32 ± 5	30 ± 4	0.08
Waist circumference (cm)	111 ± 11	113 ± 10	103 ± 11	0.003
Metabolic syndrome (%)	44%	57%	0%	0.001
ALT (IU/L)	31 ± 17	38 ± 21	25 ± 10	0.005
GGT (IU/L)	34 ± 19	38 ± 17	30 ± 18	0.05
Triglycerides (mg/dL)	182 ± 90	196 ± 103	145 ± 60	0.005
LDL cholesterol (mg/dL)	117 ± 32	106 ± 29	115 ± 30	0.4
HDL cholesterol (mg/dL)	43 ± 9	41 ± 10	45 ± 11	0.1
Metabolic syndrome	44%	57%	0%	0.001
Delta liver density–spleen density (HU)	+1 ± 7	−20 ± 4	+7 ± 6	0.001
Visceral lipids	330–463>330 ± 99	197–330	74–230	0.001

* BMI: body mass index; ALT: alanine transaminase; GGT: gamma glutamyl transferase; FL: fatty liver.

**Table 2 metabolites-13-00896-t002:** Atherosclerosis status and inflammatory biomarkers in the groups.

	Visceral Fat Storage N = 30	Liver Fat Storage N = 70	Healthy Controls N = 30	*p*-Value FL vs. Controls
Coronary soft plaques (% of total)	38	50	25	0.001
CAD: stenosis > 50% (% of total)	22	30	11	0.001
IMT *	0.86 ± 0.1	0.98 ± 0.3	0.83 ± 0.1	0.01
HOMA	3.0 ± 1.0	4.5 ± 3.0	1.5 ± 1.2	0.005
CRP(mg/dL)	0.4 ± 0.40	0.4 ± 0.3	0.3 ± 0.5	0.02
MDA(micmol/L)	69.8 ± 30	55.6 ± 35	50.0 ± 15	0.01
Paraoxonase (nmol/min)	0.2 ± 0.07	0.2 ± 0.06	0.2 ± 0.01	0.1

* IMT: intima–media thickness; CAD: coronary artery disease; HOMA: homeostatic model assessment for insulin resistance; CRP: C-reactive protein; MDA: plasma malonaldehyde.

**Table 3 metabolites-13-00896-t003:** Multivariate regression analysis for predicting coronary artery disease.

	OR *	±95% CI	*p*-Value
Diffuse fatty liver (LD-SD > −10HU)	2.75	2.3–4.9	0.001
Excessive visceral fat (>330 cm^2^)	2.01	1.2–3.8	0.001
Subcutaneous fat (>230 cm^2^)	1.15	1.0–3.6	0.2
CRP (>2 mg/dL)	1.20	0.5–1.9	0.4
HOMA (>2.5)	0.85	0.5–3.0	0.1
Metabolic syndrome (yes/no)	1.52	0.8–3.5	0.01

* OR adjusted for smoking, BMI, and age.

## Data Availability

Data available in a publicly accessible repository: The data presented in this study are openly available.

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
