# Peer review of "Liver Fat Storage Is a Better Predictor of Coronary Artery Disease than Visceral Fat"

_metabolites, 2023, doi:10.3390/metabo13080896_

Round 1
Reviewer 1 Report
The authors present experimental data to demonstrate that the metabolic syndrome is associated with hepatic and visceral fat accumulation.
The study was conducted on three groups of patients: those with excess visceral fat, then those with non-alcoholic fatty liver disease and a control group. Doppler ultrasounds were used as working techniques, with the determination of biomarkers of insulin resistance, CRP and oxidant-antioxidant status (MDA-Paraoxonase) and statistical analysis. Patients with high liver fat or high visceral fat had a higher prevalence of coronary plaques, a coronary artery stenosis and higher triglyceride levels. Multiple logistic regression analysis showed that fatty liver predicts CAD independently of visceral fat deposited Liver fat accumulation is strong in dependent risk factor for CAD and carotid atherosclerosis and contributes more than visceral fat accumulation.
I am very satisfied with the value of the work, which I consider attractive.
The abstract is well formulated.
The literature was well selected (using 13 of bibliographic references), and the critical analysis of the references is well done.
The working methods and multiple logistic regression analysis allowed a good interpretation of the obtained data.
The graphs (4) and tables (3) are very well done, and the interpretation of the experimental results corresponds to the level agreed by the journal.
The calculation relations are well established, using the known equations (The intima medial area).
Regarding the conclusions, they summarize very well the essence of the results obtained.
The bibliographic references (43) are rigorously selected from the current specialized literature.
The manuscript can be published in the journal, after the authors make corrections for the following observations: for bibliographic references, the title of the journal will be written in italics, the year of publication of the paper in bold, followed by a comma (not a semicolon), then before the pages there will be two periods (and not semicolon as it is written in most references).
Author Response
We would to thank their reviewer regarding his comments. Correction in the references were done as the author requested.
Reviewer 2 Report
In this work, the individual contribution of visceral fat and liver fat to the risk of Coronary Artery Disease is evaluated and compared. According to the data here presented, liver fat accumulation is a better predictor of CAD than visceral fat. The result is interesting, even not entirely innovative; it is not clear what is the aim of the work, whether to recommend the use of liver/spleen CT attenuation (in that case, a comparison with the state-of-art predictive tools is lacking) or to better understand the role of visceral fat and NAFLD in the pathogenesis of CAD (in that case, the role of NAFLD and visceral fat in CAD risk is not adequately discussed).
Minor issues
Reference 1 is quite old, replace it with a more recent one.
Liver fat accumulation was found to be a better predictor than visceral fat of metabolic syndrome by Lee et al (2012) [1]. This paper should be cited and discussed.
Line 200: the unit of measure for liver-spleen density is lacking and it is also lacking in table 1.
Major issues
A discussion about the traditional tools to predict CAD should be added, and the predictive power of visceral fat and fatty liver accumulation should be also compared with the traditional tools.
Also, traditional predictive tools (hypertension…) should be added to the multivariate analysis of Table 3.
In the discussion, the issue of the contribution of visceral fat to circulating FFAs and hepatic FFAs should be discussed more in detail, mentioning other relevant research papers.
1. Lee, J.; Chung, D. S.; Kang, J. H.; Yu, B. Y. Comparison of Visceral Fat and Liver Fat as Risk Factors of Metabolic Syndrome. J. Korean Med. Sci. 2012, 27, 184, doi:10.3346/JKMS.2012.27.2.184.
Author Response
- Reference 1 was replaced by new one.
- Liver fat accumulation was found to be a better predictor than visceral fat of metabolic syndrome by Lee et al (2012) [1].
This paper was discussed in lines 268-271.
- The unit of measure for liver-spleen density in line 200 was added and in table 1.
Major issues
A discussion about the traditional tools to predict CAD should be added, was added lines 305-313
In the discussion, the issue of the contribution of visceral fat to circulating FFAs and hepatic FFAs was well discussed in lines 291-308
Reviewer 3 Report
Major comment.
Most of the cited references are relatively old
Minor Comments
Line 14: “the carotid artery”. Which one? The “Common”, “Internal” or “External”?
Line 35: “(4,6)”. Does the authors mean “(4-6)”?, as there was no “5”. The references are very old.
Line 40: What is “(MAFLD)”? Is it different from “NALD”?
Line 42: Please define “MNAFLD”
Line 57: Please explain the methodology that allows the authors to determine the stage of the atherosclerosis.
Line 64: What were the BMIs of the subjects included in the study?
Line 137: The resolution of Figure 2 could be improved
Line 149: Arrows showing both visceral and superficial fat would be useful
Citations
More recent references should replace the old ones.
Reference number 1 that cited obesity is about 17 years old. A more recent one would be useful (Al Jaberi et al. (2021). Obesity: Molecular Mechanisms, Epidemiology, Complications and Pharmacotherapy. In: Tappia, P.S., Ramjiawan, B., Dhalla, N.S. (eds) Cellular and Biochemical Mechanisms of Obesity. Advances in Biochemistry in Health and Disease, vol 23. Springer, Cham. https://doi.org/10.1007/978-3-030-84763-0_13
Di Cesare et al. (2016) Trends in adult body-mass index in 200 countries from 1975 to 2014: a pooled analysis of 1698 population-based measurement studies with 19.2 million participants. The Lancet 387(10026):1377–1396. https://doi.org/10.1016/S0140-6736(16)30054-X
He et al, (2017) Prevalence of overweight and obesity in 15.8 million men aged 15–49 years in rural China from 2010 to 2014. Sci Reports 7(1):1–10. https://doi.org/10.1038/s41598-017-04135-4
Author Response
Line 14: “the carotid artery”. Common was added
Line 35: “(4,6)”. The “(4-6 is the purpose.
Line 40: What is “(MAFLD)”? The term MAFLD was dropped from paragraph
Line 42: The term MAFLD was dropped from paragraph
Line 57: Please explain the methodology that allows the authors to determine the stage of the atherosclerosis.
A: the methodology of the atherosclerosis is discussed in lines 153-169. The degree of stenosis was considered significant if >50% occlusion of the arterial lumen was present as described previously in refernces(22,23). Coronary artery disease was defined as a stenosis of >50% in at least one major coronary artery .
Line 64: What were the BMIs of the subjects included in the study?
The BMI of participants in the study is described in table 1, line 6.
Line 137: The resolution of Figure 2 was improved
Line 149: Arrows showing both visceral and superficial were added
Citations
More updated references were included. Recent references like ref 1,5 and 9 were added.
Reviewer 4 Report
In this manuscript, Liver fat accumulation is a better predictor of coronary artery disease than visceral fat was studied well. But there are some questions in the aspects of experimental designs, results and discussion and so on. Hence, I have some suggestions as follows:
1) Some descriptions in the manuscript were not exact or confusing. Some words which will make the manuscript feel like an article on a popular science book should not appear in such a research paper. The following are suggestions for improving English usage. Please use standard expression in English.
2) Please add the error analysis to every point in your tables.
3) The manuscript stays within a stage of literature survey, and is hard to find original contribution of the authors on this subject.
4) Problems on format or details: the manuscript was not well prepared according to the “Guidelines”. Please check carefully.
5) There is no clear description of experimental treatment in "Materials and Methods" and “Results”, so that it is very easy to create chaos in results. You had better transfer these general descriptions to special quantitative research.
6) All the results and discussion were listed together without logical thought.
7) If you put some photos of your experiments into the paper, the design of your experiments will be more clearly understood.
Extensive editing of English language required
Author Response
1. The manuscript was sent to specialized in English editing as the reviewer requsetd
2) Error analysis was added to our all tables.
3) Our manuscript show for the first time association between IMT, CAD and fatty liver
4. The manuscript was prepared according to the journal “Guidelines” as the reviewer requested.
5) We try to rearrange the "Materials and Methods" and “Results”.
6) The results and the discussion were re arranged
7) Figure 1 in the material and methods was added in order to explain our design
Round 2
Reviewer 2 Report
I thank the author for taking into consideration my suggestions.
Author Response
We would to thank the reviewer for his comments
Reviewer 4 Report
The manuscript has been revised and it should be checked word by word. For example, the Figure 3 should be marked A(or a) and B (or b). All the tables should be promoted according to the writing guide.
The manuscript has been revised and it should be checked word by word. For example, the Figure 3 should be marked A(or a) and B (or b). All the tables should be promoted according to the writing guide.
Author Response
The manuscript has been rearranged and rewritted